# Model Provenance Testing for Large Language Models

**Ivica Nikolić**
National University of Singapore
Singapore
inikolic@nus.edu.sg

**Teodora Baluta**
Georgia Institute of Technology
Georgia, USA
teobaluta@gatech.edu

**Prateek Saxena**
National University of Singapore
Singapore
prateeks@comp.nus.edu.sg

## Abstract

Large language models are increasingly customized through fine-tuning and other adaptations, creating challenges in enforcing licensing terms and managing downstream impacts such as protecting intellectual property or identifying vulnerabilities. We address this challenge by developing a framework for testing *model provenance*. Our approach is based on the key observation that real-world model derivations preserve significant similarities in model outputs that can be detected through statistical analysis. Using only black-box access to models, we employ multiple hypothesis testing to compare model similarities against a baseline established by unrelated models. On two comprehensive real-world benchmarks spanning models from 30M to 4B parameters and comprising over 600 models, our tester achieves $90 - 95\%$ precision and $80 - 90\%$ recall in identifying derived models. These results demonstrate the viability of systematic provenance verification in production environments even when only API access is available.

## 1 Introduction

Platforms such as Amazon SageMaker and Hugging Face have enabled wide scale distribution of ML models, most notably large language models (LLMs) [39, 47, 22]. The increase of publicly available foundation models and datasets, however, has also triggered concerns over unauthorized use of intellectual property (IP) and concerns of compromised datasets [7] or models [20]. These issues are present not just in open-source ecosystems, but also for proprietary models that are hidden behind APIs [3, 32]. For instance, concerns about model stealing attacks wherein one can extract the model parameters even for production-level models are on the rise [8, 42, 33]. Similarly, there is growing concern that models may contain backdoors or vulnerabilities, making them susceptible to jailbreaking [20, 2, 59]. Despite best efforts to create a safe environment for the development of foundation models, there have already been instances of reported misuse [38, 16, 41].

This landscape highlights the growing need for *model provenance testing*. The problem of model provenance testing is as follows: Determine whether a target model has been derived from a foundational model by customizations such as fine-tuning. This problem has applications in tracking reuse of models not just in open marketplaces but also across product teams in large organizations. When a security or privacy audit finds a problem with a foundational model, it becomes important to identify which other models in use by the organization may be derived from the problematic one and take appropriate remedial actions (e.g. revoke, retrain, or fortify) to mitigate the risk of further

non-compliant use. Model provenance tracking is often useful after the fine-tuned model has been deployed, and when authentic ground truth is unavailable or unreliable.

One challenging aspect of designing a model provenance tester is achieving high accuracy. There is a cost associated with a provenance verdict. For instance, as a result of provenance tracking, a company may initiate legal action or investigation. For use cases within the same organization, developers might have to revoke the use of an existing model and even retrain from a clean parent model. False positives, i.e., the deployed LLM is wrongly flagged as a derivation of a problematic LLM, thus entail a downstream cost. At the same time, false negatives, i.e., not being able to flag that the LLM is customized from a problematic parent, also increase the risk of non-compliance. Therefore, we want a principled way to decide provenance and to make accuracy trade-offs.

Another challenge is that a practical provenance tool needs to have *minimal assumptions* to be readily usable in many post-deployment settings. We focus on techniques that do not change typical training and data pipelines, and can be integrated for current state-of-the-art LLMs. The tester is expected to only have *black-box query* access to the models and has no additional information, such as the training dataset, test set, or the algorithm used for training. We are not aware of any prior work addressing the question of model provenance testing systematically and in such practical setups.

**Contributions..** In this paper, we design the first practical model provenance tester for LLMs that requires only query access. Our proposed techniques stem from a key empirical observation: The output distribution of fine-tuned models is often close to that of their parent model. This distance between a model and its true parent is typically smaller than that between the model and other unrelated models, making it possible to reliably trace back a derived model to the original parent. In order to keep assumptions to a minimum, we propose to employ the principled framework of statistical hypothesis testing. Specifically, we use black-box sampling and estimation to determine whether the distribution captured by the given model is close to that of the candidate parent. Such estimation can provide formal statistical significance measures, which can be used to check for the *null hypothesis*, i.e., the customized LLM is not close to the given parent model. Our approach is agnostic to the chosen metric for closeness and returns with the guarantee that the false positive rate is less than or equal to a user-specified threshold. We conduct an extensive empirical evaluation across two comprehensive benchmarks comprising over 600 models from Hugging Face, ranging from 30M to 4B model parameters and across diverse domains. Our tester achieves 90-95% precision and 80-90% recall in detecting model provenance, even with a limited number of queries.

## 2    Model Provenance Testing

Pretraining LLMs involves significant investment, requiring substantial computational resources costing millions of dollars in infrastructure and thousands of GPU hours. When Company A releases a pretrained LLM denoted as $f$, it employs specific licensing terms crucial for protecting this investment, maintaining competitive advantage, and controlling the model's usage [32, 3]. Startup B might download $f$, perform only fine-tuning or other light customization (mixture-of-experts, prompt engineering), but claim to have pretrained their model $g$ from scratch, thereby circumventing licensing requirements and misrepresenting company A's work. In such cases, we want to be able to *determine if $g$ is derived through fine-tuning of $f$* and resolve the *model provenance problem*.

We consider model provenance testing framework with *minimal assumptions* only, where the tester can query the models $f, g$ on arbitrary prompts (e.g., through an API) and get responses. The tester has no access to the training datasets, embeddings, hyperparameters used by either company, or any information about potential modifications performed by Startup B. This mirrors real-world conditions where companies do not always disclose their training procedures, data sources, or modification techniques, making the provenance testing problem both practical and challenging.

Our proposed methods are evaluated for a *non-adaptive adversary*, i.e., the Startup B is not aware of the strategy of the model provenance tester. On the other hand, it is not clear if more advanced techniques for evasion of provenance detection deployed by *adaptive adversaries* can provide real benefit due to the randomness used in our tester (sampling from infinite set of prompts) and the fact that adversarial fine-tuning may come at the cost of reduced model performance on intended task.

## 2.1 Related Work

The problem of testing provenance is similar to designing schemes to prove ownership of a model. We provide more details on related work in Appendix A and summarize main approaches here.

**Fingerprinting.** Fingerprinting aims to create an identifier of the model for different downstream purposes such as IP protection. Most prior work require white-box access or additional knowledge such as intermediate LLM modules [55], access to training and testing datasets and changes to the fine-tuning [49]. One type of approaches detect copies or tampering by hashing the model weights [9, 48]. Other works aim to create fingerprints by estimating the decision boundary via adversarial examples [27, 50, 37]. However, adversarial-based approaches can still have a high false positive rate, i.e., two unrelated models have the same fingerprinting samples [6]. Follow-up work generates universal perturbations that can be added to samples to match the target model [27]. This can be prone to adversarial evasion [19]. Thus, recent work constructs a set of adversarial examples that are connected with each other and that characterize the decision boundary more robustly [50]. In addition to model fingerprinting via adversarial perturbations, there are works that focus on measuring the model similarity [19, 17, 11, 23, 14]. For example, one work selects samples with different prediction results across two groups of reference models [17]. This approach utilizes their pairwise correlation to identify target stolen image classification models. Other approaches also rely on the internal representations to compare models, without focusing on provenance. For instance, one work locally approximates the models under test with linear models and then computes the cosine distance between the weights of the linear model [23]. One recent work in black-box access requires specialized prompting and an additional model's embeddings to identify models [34]. We highlight that these approaches do not focus on the question of model provenance as fingerprints may not be robust to fine-tuning and other customizations. Our tester also has minimal assumptions of black-box access to only next-token predictions. While our work considers this measure of similarity, our approach returns with a bounded false positive rate, which is metric-agnostic.

**Watermarking.** Other approaches prove ownership via embedding watermarks in the model weights [10, 45, 43], hidden-layer activations [13] or other model features [26, 12]. Another type of watermarks are achieved by learning triggers or backdoors that produce predefined outputs [24, 40, 52, 36, 6, 56]. These rely on implementing changes to the white-box model, fine-tuning or training the model on a specialized dataset and can affect the model's performance. Specialized model ownership schemes have also been proposed for other models such as graph neural networks [58, 44]. Most existing approaches do not have provable guarantees that model ownership can be verified with a given confidence, so the verification is often empirically determined. It is particularly challenging to preserve the watermark detection after applying model customizations [25].

The focus of prior work has not been on designing tests for determining provenance under model customizations and do not work under the same minimal assumptions. For instance, recent work proposed a detection framework for fine-tuning and pruning that still requires white-box access to the victim model to generate test cases and it does not consider LLMs [11]. Moreover, our benchmarks are much more extensive with hundreds of models under diverse customizations.

## 3 Approach

Our approach to testing model provenance is based on a key observation: fine-tuning and other model derivation techniques typically result in only limited changes to the original model, as they primarily adapt the model for new tasks. After fine-tuning, the derived model $g$ may remain similar to its parent model $f$, as the process focuses on refining specific capabilities rather than creating fundamental changes to the model distribution.

## 3.1 Model Provenance Tester

Since the tester has only query access to the models, the only data it can collect is from providing inputs (called *prompts*) and analyzing the corresponding output tokens. Furthermore, due to our minimal assumptions and lack of information about the training datasets, the tester queries the models on randomly chosen short sequences, further denoted as a prompt space $\Omega$.

The tester independently samples from $\Omega$ a set of $T$ prompts $x_1, \ldots, x_T$, queries each model on the same set of prompts, and then compares their output tokens pairwise. For each prompt, we compare only the first output token generated by each model; however, $n$-grams could also be considered. The *similarity* between two models $f$ and $g$ is then calculated as the proportion $\mu$ of prompts on which the models produce the same output token: $\mu = \frac{1}{T} \sum_{j=1}^{T} \mathbb{1}(f(x_j) = g(x_j))$.

However, the ratio $\mu$ by itself may not be sufficient to determine whether $f$ and $g$ are similar, since even two unrelated models might agree on some proportion of outputs by chance. To assess the significance of $\mu$, we introduce a set of *control models* $C = \{c_1, \ldots, c_m\}$ for the model $f$ and compare their similarities to $g$. These control models are sampled from the set of available LLMs with distinct architectures or training data, to ensure diversity. Our aim is to compare $\mu$ to the similarity ratios $\mu_i$ between each control model $c_i$ and $g$. The number $m$ of control models and their sampling strategy depend on availability; generally, the larger the set of control models, the better the estimation of similarity among unrelated models.

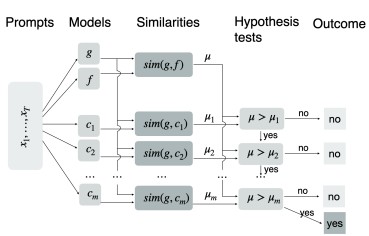

Figure 1: Our model provenance tester that decides if model $g$ is derived from model $f$.

The final step of the tester is to verify that the similarity ratio $\mu$ between $f$ and $g$ exceeds all similarity ratios $\mu_i$. However, we want to ensure this is not merely due to random chance, but rather reflects a true difference. To establish such theoretical guarantees, we employ multiple hypothesis testing. More precisely, for each control model $c_i$, we formulate the following hypothesis test $H^i$:

$$H_0^i: \quad \mu \leq \mu_i,$$
$$H_1^i: \quad \mu > \mu_i,$$

where $H_0^i$ is the null hypothesis that the similarity between $f$ and $g$ is less than or equal to the similarity between $c_i$ and $g$, and $H_1^i$ is the alternative hypothesis that the similarity between $f$ and $g$ is greater. To test each of the hypothesis $H^i$, we employ a z-test, which is a standard statistical test well-suited for comparing proportions like our similarity ratios when working with large samples. The z-test produces a $p$-value, which represents the probability of observing such a difference in proportions if there were truly no difference between the models (i.e., if the null hypothesis were true). A small $p$-value (typically less than a predetermined significance level $\alpha$) indicates that the similarity $\mu$ between $f$ and $g$ is indeed significantly larger than the similarity $\mu_i$ between $g$ and $c_i$.

When conducting multiple hypothesis tests simultaneously, we want to maintain this same overall risk level of $\alpha$, regardless of how many tests we perform. However, running multiple tests increases our chances of obtaining at least one false positive across all tests, known as the family-wise error rate (FWER). To control this cumulative risk, we employ the Holm-Bonferroni method [18], which adjusts the significance thresholds $\alpha_k$ for individual tests $H^k$ to ensure the overall false positive rate remains at or below our desired level of $\alpha$. We illustrate this basic tester idea in Figure 1.

The provenance tester described thus far assumes we have one candidate parent $f$ that we want to test our child model $g$ against. We now extend it to the general case where the goal is to determine whether a model $g$ is derived from some model from the set $f_1, \ldots, f_s$ of candidate parent models. While running the basic tester $s$ times (once for each provenance pair $(f_i, g)$) would solve the unspecified parent problem, it would also would require additional correction for multiple testing to maintain the same level of confidence. The probability of false positives would grow with the number of candidate parents $s$ unless appropriate adjustments are made to the significance level.

We thus propose the improved tester in Algorithm 1. Our tester avoids this issue by conducting a single set of hypothesis tests after identifying the most similar candidate. First, it finds the most similar model to the given model $g$ among all the control models $C$ and candidate parents $F$. If that model is a control model, the algorithm terminates with False. Otherwise, it goes on to test whether the FWER of this model is overall below $\alpha$, the desired significance level. When this test procedure returns False, it indicates that we could not establish this higher similarity with the desired level of statistical significance. This may occur either because there is genuinely no significant similarity indicative of provenance or because the test lacked sufficient power under the given parameters (e.g.,

**Algorithm 1** Provenance Tester for $g$ Given a Candidate Parent Set

---

**Require:** Model $g$, candidate set $F = \{f_1, \ldots, f_s\}$, set of control models $C = \{c_1, \ldots, c_m\}$, prompt space $\Omega$, number of prompts $T$, significance parameter $\alpha$, statistical test ZTest.

$\quad x_1, \ldots, x_T \overset{\text{iid}}{\sim} \Omega$ $\hfill \triangleright$ Sample T prompts

$\quad$ **for** $i \leftarrow 1$ to $s$ **do**

$\qquad \mu_i \leftarrow \frac{1}{T} \sum_{j=1}^{T} \mathbb{1}(f_i(x_j) = g(x_j))$ $\hfill \triangleright$ Calc sim of candidates

$\quad$ **end for**

$\quad$ **for** $i \leftarrow 1$ to $m$ **do**

$\qquad \mu_i' \leftarrow \frac{1}{T} \sum_{j=1}^{T} \mathbb{1}(c_i(x_j) = g(x_j))$ $\hfill \triangleright$ Calc sim of controls

$\quad$ **end for**

$\quad \mathcal{M} \leftarrow \{\mu_1, \ldots, \mu_s\} \cup \{\mu_1', \ldots, \mu_m'\}$ $\hfill \triangleright$ Set of all sims

$\quad \mu_{max} \leftarrow \max(\mathcal{M})$ $\hfill \triangleright$ Find highest sim

$\quad$ **if** $\mu_{max} \notin \{\mu_1, \ldots, \mu_s\}$ **then return** FALSE $\hfill \triangleright$ Highest not from $F$, but from $C$, so cannot be parent

$\quad$ **end if**

$\quad$ **for** $\mu' \in \mathcal{M} \setminus \{\mu_{max}\}$ **do**

$\qquad p_i \leftarrow \text{ZTest}(\mu_{max}, \mu', T)$ $\hfill \triangleright$ Compare against other sims

$\quad$ **end for**

$\quad (p_{(1)}, \ldots, p_{(s+m-1)}) \leftarrow \text{Sort}(p_1, \ldots, p_{s+m-1})$

$\quad$ **for** $k \leftarrow 1$ to $s + m - 1$ **do**

$\qquad \alpha_k \leftarrow \alpha/(s + m - k)$ $\hfill \triangleright$ Holm-Bonferroni adjustment

$\qquad$ **if** $p_{(k)} > \alpha_k$ **then return** FALSE

$\qquad$ **end if**

$\quad$ **end for**

$\qquad$ **return** $(\text{TRUE}, \arg\max_{i \in [s]} \mu_i)$ $\hfill \triangleright$ Return parent

---

sample size or number of prompts) to detect it. When the algorithm return True, it has the guarantee that the most similar model is one of the candidate models and that the total significance level across all hypotheses meets the threshold $\alpha$.

## 3.2 Understanding Sources of Error

The overall error rates of our tester depend on the combination of errors from both our statistical hypothesis testing and the two core assumptions which we describe below.

**Assumption 1: Derivation implies similarity.** We assume that when model $g$ is derived from $f$, they will exhibit above-average similarity in their outputs. This assumption leads to two potential types of errors: false negatives and false positives. False negatives occur when a derived model shows insufficient similarity to its parent. This can happen when a model customizer applies extensive modifications that significantly alter the model's behavior. While resource constraints typically prevent such extreme modifications (as they would approach the cost of training from scratch), some legitimate derivation relationships may still go undetected. False positives arise when independently developed models exhibit high similarity. This typically happens when models are trained on similar datasets or designed for similar specialized tasks - for instance, two independent medical diagnosis models may produce very similar outputs due to their shared domain constraints.

**Assumption 2: Control models establish a valid baseline.** We assume our control models provide a reliable baseline for the similarity we should expect between unrelated models. Similarly, poor selection of control models can lead to two types of errors. False positives occur when our control models are too dissimilar from the domain of $f$. For example, using general language models as controls for specialized code generation models sets an artificially low baseline, making normal domain-specific similarities appear significant. False negatives happen when control models are themselves derived from $f$ or trained on very similar data. This establishes an artificially high baseline that masks genuine derivation relationships.

While we can provide theoretical guarantees for controlling error rates in hypothesis testing, we cannot derive analytical bounds for errors arising from the assumptions about derivation implying

similarity or the validity of the control model baseline. The assumption-based error rates can only be evaluated empirically. We find these hold in practice through extensive experiments (Section 4).

### 3.3 Reducing Query Complexity

Most of LLMs available currently allow cheap (even free) API access, thus the monetary query cost of running our testers is insignificant. However, sometimes this is not the case, for instance when the cost of queries is high (e.g. one query to OpenAI model O1 can cost more than \$1 [31]), or when the models have some rate restrictions. Furthermore, there are use cases when query complexity can be reduced without any side effects, thus it makes sense from optimization perspective.

We can divide the queries used in the tester into two distinctive types: *online queries* made to the tested child model $g$, and *offline queries* made to the parent model $f$ (or models $f_1, \ldots, f_s$) and to the control models $c_1, \ldots, c_m$. We make this distinction for two reasons. First, often offline queries are much cheaper, as the potential parent models (and the control models as we will see in the Section 4) are well established, and available from multiple sources, thus they are usually cheaper or free. Second, in some use cases, we can reuse the offline queries to perform many provenance tests of different $g_i$. We analyze separately these two scenarios.

**Reducing Online Complexity.** Since our tester is fundamentally based on statistical hypothesis testing, any reduction in query complexity must be compensated by increasing the statistical power of individual queries. Rather than querying model $g$ with $T$ random prompts, we can strategically select a smaller set of $T' < T$ prompts that yield comparable statistical power for detecting model provenance[1]. We achieve this through an informed sampling approach: instead of uniform sampling from $\Omega$, we employ rejection sampling with an entropy-based selection criterion. Specifically, to generate each prompt in $T'$, we sample $k$ candidate prompts from $\Omega$ and select the one that maximizes the entropy of output tokens across all parent and control models. The selection criterion is dynamically weighted to favor prompts that have stronger discriminative power between similar models. While this approach introduces dependencies between the sampled prompts (so the theoretical guarantees of hypothesis testing used in Algorithm 1 do not carry over), our empirical results in Section 4.4 demonstrate its practical effectiveness. We detail our approach in Appendix H.

**Reducing Offline Complexity.** We further consider the case of reducing offline complexity in settings where offline queries cannot be reused. The key observation for reducing offline query complexity is that we may not need an equal number of queries to all parent/control models to identify the most similar one. If a particular parent model shows consistently higher similarity to $g$ compared to other models, we might be able to confirm it as the top candidate with fewer queries to the clearly dissimilar models. The challenge lies in determining when we have sufficient statistical evidence to conclude that one model is significantly more similar than the others, while maintaining our desired confidence levels. This observation naturally leads us to formulate the problem as a Best Arm Identification (BAI) [5] problem in the Multi-Armed Bandit (MAB) setting. In this formulation, each parent or control model represents an "arm" of the bandit, and querying a model with a prompt corresponds to "pulling" that arm. The "reward" for each pull is the binary outcome indicating whether the model's output matches that of the tested model $g$. The goal is to identify the arm (model) with the highest expected reward (similarity to $g$) while minimizing the total number of pulls (queries). We implement an optimization for our tester based on state-of-the-art BAI approaches (see Appendix I for details) and show it impact in Section 4.4.

## 4 Evaluation

We evaluate our proposed provenance testing approach experimentally and seek to answer the following research questions:

(RQ1) How accurate is our provenance tester in practice and how does the number of prompts affect its performance?

(RQ2) To what extent do derived models maintain similarity to their parents?

---

[1]It means in Algorithm 1, instead of random sampling $x_1, \ldots, x_T \overset{\text{iid}}{\sim} \Omega$, the goal is to find set $x_1, \ldots, x_{T'}$ from $x_1, \ldots, x_T$ and $F, C$.

(RQ3) How does the size and selection of control models impact the tester?

(RQ4) How effective are the query reduction approaches?

## 4.1 Models and Provenance Testing Parameters

We collect model candidates for all provenance pairs from the Hugging Face (HF) platform [21]. To avoid selection bias, we used download counts as our selection criterion, taking the most popular models subject only to hardware constraints on model size.

To increase variety of candidates, we create two distinct benchmarks BENCH-A and BENCH-B, that differ in aspects such as model sizes, choice of pre-trained models, and ground-truth verification procedure (refer to Tbl. 1). The full procedure of collection of models and constructions of benchmarks is described in Appendix C. We use the standard significance $\alpha = 0.05$ (see Appendix G for other values). Sampling of prompts is given in Appendix D.

The implementation of the tester along with the two benchmarks can be found at `https://github.com/ivicanikolicsg/model_provenance_testing`.

**Selection of control set.** In all of our provenance tests, we use the complete set of available pre-trained models from the benchmark as control models - 10 models for BENCH-A and 57 for BENCH-B. This selection was done to demonstrates that effective control sets can be constructed without careful manual curation or domain-specific analysis. Manual curation would not have been feasible since our two benchmarks have over 600 candidates. Specifically, we make no effort to align control models with particular parent models' domains or capabilities. We neither analyze the outputs of parent models $f$ nor attempt to match control models to specific use cases. Instead, we simply include all pre-trained models that rank among the most popular on the Hugging Face platform. This sampling approach, while simple, helps avoid introducing obvious selection bias while ensuring broad coverage of model types and capabilities. Due to the large benchmark size, we find diverse examples of candidates for domains such as financial, medical and more (see Appendix C for details). We have not evaluated other selection strategies, which remains a direction for future work. This straightforward selection strategy, however, allows us to evaluate whether provenance testing can be effective even without carefully chosen control sets.

Table 1: Comparison of BENCH-A to BENCH-B on different features.

| Feature | BENCH-A | BENCH-B |
|---|---|---|
| pre-trained models | 10 | 57 |
| derived models | 100 | 383 |
| total models | 100 | 531 |
| model parameters | 1B-4B | $< 1$B |
| compilation method | manual | automatic |
| ground-truth verification | higher | lower |

## 4.2 Accuracy of Model Provenance Tester

We evaluate the accuracy of the provenance tester by examining its performance on both BENCH-A and BENCH-B under different numbers of prompts. Figure 3 shows the precision and recall results from these experiments.

The precision is notably high (approximately $0.95$) when the tester uses up to $1,000$, and it is significantly higher than the baseline precision of randomly guessing the parent which is $\frac{1}{10+1} \approx 0.09$ for BENCH-A and $\frac{1}{57+1} \approx 0.02$ for BENCH-B. Interestingly, however, the precision reduces as the number of prompts (test samples) increases. This is in direct contrast to common hypothesis testing, where larger sample size leads to smaller standard errors, thus higher precision. We get different results because our model provenance tester relies on detecting similarities of models. When using a smaller number of prompts, it can detect only the stronger similarities which are usually due to model provenance. However, as we increase the prompts, it starts detecting similar models that not necessarily have provenance relation. This leads to misclassification and reduced precision.

The recall behavior shows an opposite trend - it improves with a larger number of prompts, eventually reaching $80\% - 90\%$ depending on the benchmark. This follows expected behavior: more prompts increase the statistical power of our hypothesis tests, enabling detection of small but significant differences in similarities. This increased sensitivity leads to higher recall rates, as the tester can detect more subtle provenance relationships that might be missed with fewer prompts.

We also examine the impact the randomness of prompt sampling on the tester's accuracy. We conduct experiments on both benchmarks using five different randomly sampled sets of $1,000$ prompts, with the same set of prompts used in all testers, and record the precision and recall for each run – see Table 6 of Appendix E. The results show that these values vary by $1 - 4\%$ between runs, indicating consistent performance across different prompt samples.

> **(RQ1):** *Our model provenance tester demonstrates high accuracy across different benchmarks, achieving precision of $90\% - 95\%$ and recall of $80\% - 90\%$ with $3,000$ prompts per model. Simply increasing the number of prompts does not guarantee uniformly better results, reflecting a fundamental trade-off: gains in recall might be accompanied by losses in precision.*

### 4.3 Correctness of Assumptions

As discussed in Section 3.2, our approach relies on two key assumptions. While the high accuracy demonstrated in the previous section indirectly validates these assumptions, we provide here a detailed experimental analysis of both.

Our first assumption posits that derived models maintain significant similarity to their parent models. To evaluate this, we analyzed the similarity rankings across all provenance tests using $3,000$ prompts. For each derived model, we examined where its true parent ranked among all models in terms of similarity ratio $\mu$. The results strongly support this: in BENCH-A, the true parent had the highest similarity ratio in $93\%$ of cases, while in BENCH-B this occurred in $89\%$ of cases. When considering whether parents ranked in the top $50$th percentile by similarity, these percentages increased to $98\%$ and $96\%$ respectively. Thus we can conclude that our experiments indicate that derived models do indeed maintain strong similarity patterns with their parent models. Inadvertently, we have shown as well that with $3,000$ prompts the tester almost approaches the statistical limit (only the model with highest similarity ratio can be identified as a parent), as the recalls are very close to the percentages of highest similarity ($89\%$ recall vs. $93\%$ highest parent

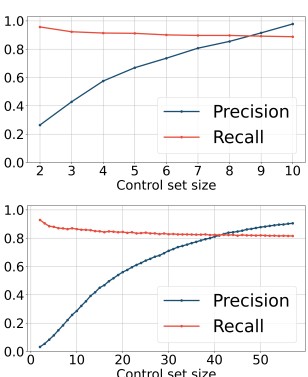

Figure 2: Precision and recall of BENCH-B (left) and BENCH-A (right) with respect to smaller control set size.

similarity, and $82\%$ recall vs. $89\%$ similarity, for the two benchmarks, respectively).

> **(RQ2):** *The assumption of similarity between derived and parent models is valid for most provenance pairs.*

Our second assumption concerns whether control models can effectively establish a baseline for similarity between unrelated models. We stress that in our experiments we have chosen the control models to be simply the set of all pre-trained models in an unbiased way, without any special selection or optimization for particular parent models they are tested against. We empirically observe that such unbiased selection of control model establishes a good baseline similarity as evident from the accuracy results presented thus far.

We further examine how the size and quality of the set of control models might affect tester accuracy. We conducted experiments varying the size of the control set while keeping other parameters constant ($3,000$ prompts per test). We randomly sampled different-sized subsets from our full control sets (10 models for BENCH-A and 57 for BENCH-B) and ran 100 complete benchmark tests for each size, and averaged the outcomes. The results, shown in Figure 2, reveal that both size and quality of the control set significantly impact tester performance. For BENCH-A, even with just $4$ control models, the tester achieved $55\%$ precision. This relatively good performance with few controls can

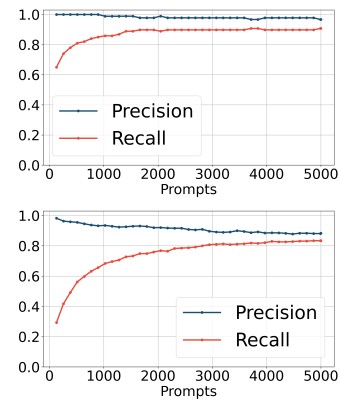

Figure 3: Precision and recall of the model provenance tester with different number of prompts on BENCH-A (top) and BENCH-B (bottom).

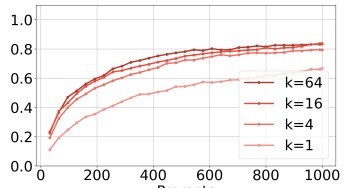 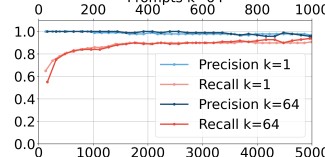 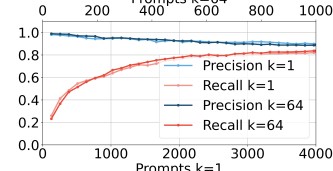

Figure 4: Recall for BENCH-B with different values of advanced prompt sampling ($k$).

Figure 5: Precision/recall for BENCH-B (left) and BENCH-B (right) when advanced online prompt sampling with $k = 64$ uses four times less prompts than no advanced sampling ($k = 1$).

be attributed to BENCH-A consisting entirely of general-purpose, well-trained LLMs - thus any subset of these models provides a reasonable baseline for measuring similarity between unrelated models. However, for BENCH-B, the randomly sampled 4-model control set yielded less than $10\%$ precision. This poor performance stems from BENCH-B containing a much more diverse set of models, including domain-specific models (e.g., for medical or coding tasks) and smaller models with varying capabilities. With such diversity, a small random subset of control models is unlikely to establish good baselines for all test cases - for instance, when testing a coding-focused model, we need coding-related models in the control set to establish proper baselines[2]. Performance improves steadily as control set size increases in both benchmarks, since larger control sets are more likely to include appropriate baseline models for each test case.

---

*(RQ3): The tester's performance degrades when the control set is too small or poorly selected.*

---

### 4.4   Reducing Query Complexity

To reduce the online complexity, we implement an advanced rejection prompt sampling strategy as detailed in Section 3.3. We evaluate this strategy using different parameter values $k = 4, 16$, and $64$ (recall, $k$ defines how many random samples are used to produce one selected sample), comparing it to the standard provenance testing without rejection ($k = 1$).

Figure 4 compares the tester's recall across different values of $k$. Notable improvements are visible even at $k = 4$, with higher values of $k$ showing better results (though with diminishing returns). Specifically, the recall achieved with $1,000$ prompts at $k = 1$ can be matched using only about $250$ prompts at $k = 64$, representing a four-fold reduction in online complexity. Figure 5 provides a comprehensive comparison between $k = 1$ and $k = 64$ for both precision and recall across both benchmarks, using $4 - 5$ times fewer queries for $k = 64$ (note, in Figure 5 the number of prompts for $k = 64$ are given at the top of the plots). The results demonstrate that the tester maintains its effectiveness despite the significant reduction in queries to the tested models. For example, advanced prompt sampling achieves high

Table 2: Precision and recall of the base vs BAI tester on BENCH-A and BENCH-B.

| Allowed Queries $T$ | Benchmark | Tester | Avg Queries | Precision | Recall |
|---|---|---|---|---|---|
| 500 | BENCH-A | base | 500 | 1.00 | 0.81 |
| 500 | BENCH-A | BAI | 450 | 0.98 | 0.29 |
| 500 | BENCH-B | base | 500 | 0.95 | 0.56 |
| 500 | BENCH-B | BAI | 452 | 0.98 | 0.29 |
| 1,000 | BENCH-A | base | 1,000 | 0.99 | 0.86 |
| 1,000 | BENCH-A | BAI | 605 | 1.00 | 0.63 |
| 1,000 | BENCH-B | base | 1,000 | 0.94 | 0.68 |
| 1,000 | BENCH-B | BAI | 809 | 0.98 | 0.42 |
| 2,000 | BENCH-A | base | 2,000 | 0.98 | 0.89 |
| 2,000 | BENCH-A | BAI | 1,482 | 0.97 | 0.54 |
| 2,000 | BENCH-B | base | 2,000 | 0.92 | 0.77 |
| 2,000 | BENCH-B | BAI | 1,482 | 0.97 | 0.54 |

levels of $90 - 95\%$ precision and $80 - 90\%$ recall while reducing the required number of prompts from $3,000$ to just $500$ per model.

We next evaluate strategies for reducing offline complexity, which refers to the number of queries made to pre-trained models during testing. We implement this reduction using BAI, as described in Section 3.3 and given in Algorithm 4. We test this approach on both benchmarks by setting a target budget of $T$ queries (prompts) per pre-trained model. For example, with $T = 1000$ on BENCH-A,

---

[2]Note that in practice, unlike our random sampling experiments, one can deliberately select control models matching the domain and capabilities of the suspected parent model, thus reducing significantly the impact of size of control sets, and leaving quality of the control set as the main factor on efficiency of the tester.

which contains 10 pre-trained models, the BAI-based provenance tester has a maximum budget of $10 \cdot 1,000 = 10,000$ total queries to make its decision.

Table 2 compares the performance of the base tester and the BAI-enhanced version across different query budgets $T \in \{500, 1000, 2000\}$. The results show that the BAI tester successfully reduces offline complexity by $10\% - 30\%$ (as shown in the "avg queries" column). However, this reduction comes at a significant cost to recall, while precision remains largely unchanged. For instance, with $T = 1,000$ on BENCH-A, BAI reduces the average number of queries from $1,000$ to $605$, but recall drops from $0.86$ to $0.63$. Similarly, on BENCH-B, the average queries decrease from $1,000$ to $809$, but recall falls from $0.68$ to $0.42$. This pattern persists across different values of $T$ and both benchmarks, suggesting that the trade-off between query reduction and recall preservation is not favorable in most cases.

> **(RQ4)**: *The online query optimization strategy leads to a 4-6 fold query reduction without accuracy drop, whereas the offline approach performs only marginally better and has a negative impact on recall.*

## 5 Conclusion

Our work formulates the model provenance testing problem for LLMs which has many applications such as in detection of misuse of terms of use or vulnerable customized models. We present an approach based on statistical testing with minimal assumptions that has high accuracy for real-world benchmarks. Our key insight is that models derived through standard customization approaches maintain a level of similarity to their parent model that is statistically distinguishable from unrelated parents. We evaluate this observation empirically, together with our approach and several optimizations. We find our proposed method to be practical for deciding LLM provenance.

## Acknowledgments

We thank Xu Louise, Bo Wang, Jason Zhijingcheng Yu, and Mallika Prabhakar for their valuable feedback on the draft. This research is supported by a Singapore Ministry of Education (MOE) Tier 2 grant MOE-T2EP20124-0007. This material is based upon work supported by the U.S. National Science Foundation under Grant No. 2531010. Any opinions, findings, and conclusions or recommendations expressed in this material are those of the authors and do not necessarily reflect the views of the NSF.

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

**Algorithm 2** Model Provenance Tester for Pair $(f, g)$
---
**Require:** Pair $(f, g)$, set of control models $C = \{c_1, \ldots, c_m\}$, prompt space $\Omega$, number of prompts $T$, significance parameter $\alpha$, statistical test ZTest.

$\quad x_1, \ldots, x_T \overset{\text{iid}}{\sim} \Omega$ $\hfill \triangleright$ Sample T prompts

$\quad \mu \leftarrow \frac{1}{T} \sum_{j=1}^{T} \mathbb{1}(f(x_j) = g(x_j))$ $\hfill \triangleright$ Calc sim of f and g

$\quad$ **for** $i \leftarrow 1$ to $m$ **do**

$\quad\quad \mu_i \leftarrow \frac{1}{T} \sum_{j=1}^{T} \mathbb{1}(c_i(x_j) = g(x_j))$ $\hfill \triangleright$ Calc sim of $c_i$ and $g$

$\quad\quad p_i \leftarrow \text{ZTest}(\mu, \mu_i, T)$ $\hfill \triangleright$ Obtain p-values

$\quad$ **end for**

$\quad (p_{(1)}, \ldots, p_{(m)}) \leftarrow \text{Sort}(p_1, \ldots, p_m)$ $\hfill \triangleright$ Sort p-values

$\quad$ **for** $k \leftarrow 1$ to $m$ **do**

$\quad\quad \alpha_k \leftarrow \alpha/(m - k + 1)$ $\hfill \triangleright$ Holm-Bonferroni adjustment

$\quad\quad$ **if** $p_{(k)} > \alpha_k$ **then return** FALSE $\hfill \triangleright$ Not a provenance pair

$\quad\quad$ **end if**

$\quad$ **end for**

$\quad\quad$ **return** TRUE $\hfill \triangleright$ Is a provenance pair
---

# 6   Appendix

# A   Related Work

We further detail prior work's required access to the model and any additional knowledge to identify or show ownership of a model, in particular large language models in Table 3.

White-box setting is the most prevalent in existing works. In particular, it is assumed that the analysis is done by the model owner or defender who has access to its model weights. Our approach is fully black-box based and it only analyzes the first output token of each query. For instance, in [53], the defender has access to their own model's parameters and the last linear layer as a fingerprint, and aims to verify ownership through API access to the suspect model. LLMmap considers only different versions of the same model and 5 model variants that have resulted from fine-tuning [34]. LLMmap proposes to train a model on a set of queries to detect unusual responses from a model which may come in response to an attacker's fingerprinting probing. It uses a pre-trained textual embedding model to generate vector representations of the query and responses that they then use to classify and identify models. It also uses heuristics to identify the most promising prompts for model identification. Other LLM fingerprinting approaches uses system-level statistics to analyze the fingerprint models such as inter-token times when these are available either locally or over the network [1].

# B   Basic Tester

A pseudo-code of the basic tester is given in Algorithm 2. In summary, when the procedure returns True, it ensures that the total family-wise error rate (FWER) is controlled at the significance level of $\alpha = 0.05$ or lower. This means we can confidently state that the similarity between $f$ and $g$ is significantly higher than between $g$ and any control model, supporting the existence of a provenance relationship. Conversely, when the procedure returns False, it indicates that we could not establish this higher similarity with the desired level of statistical significance. This may occur either because there is genuinely no significant similarity indicative of provenance or because the test lacked sufficient power under the given parameters (e.g., sample size or number of prompts) to detect it.

# C   Models and Benchmarks

We collect model candidates for all provenance pairs from the Hugging Face (HF) platform [21]. Since there is no inherent ground truth to determine whether two models constitute a provenance pair, we employ multiple heuristic approaches. These include analyzing metadata available through the HF API and comparing the weights of downloaded models. We consider the most reliable ground

Table 3: Comparison of our work to related work based on fingerprinting.

| Work | Access Type | Technique | Additional Knowledge |
|------|-------------|-----------|----------------------|
| Our | Black-box with only the next token | Hypothesis testing with statistical guarantees | No |
| [34] | Black-box with logits | Proposes to train a model on a set of queries | Pre-trained embeddings for classifier; Queries manual crafted and domain knowledge |
| [57] | White-box model internal representations | Classifiers trained on representations | No |
| [51] | White-box. Models are embedded with fingerprint key pairs | Embedding robust fingerprints into the owner model. | Changes in training procedure |
| [1] | System-level statistics such as inter-token times | Deep learning to classify | Network traffic |
| [4] | Same as LLMmap | LLMmap and other dynamic fingerprinting | Judge model, additional access to classifier models and manual fingerprinting |
| [53] | White-box. Model's parameters and the last linear layer as a fingerprint and API access to the suspect model. | Vector similarity | No |
| [29] | Takes datasets of AI-generated texts and classifies them in families | Linguistic features for similar styles | No |
| [54] | White-box access to base model, black-box access to the suspect model | A proactive fine-tuning approach to modify the output of the model | No |
| [28] | Knowledge of the training dataset | Statistical testing and distance to margin | Classification models |
| [30] | Log-probabilities of the target and random perturbations of the input passage to another model | Used to decide machine-generated text | Additional model |

truth to be cases where model uploaders explicitly specify their model as a fine-tuned version of another model, indicated by the presence of `"base_model:finetune:<basemodel_name>"` keyword in the model description on HF. When this explicit indication is not present, we resort to less reliable methods: we attempt to infer parent-child relationships through model naming patterns and by analyzing model descriptions on HF. Additionally, we identify potential provenance pairs by measuring the similarity between model weights, assuming that highly similar weights suggest a parent-child relationship. From these models we build two benchmarks.

The first benchmark, called BENCH-A, consists of LLM pairs for model provenance constructed from popular pre-trained models and their fine-tuned derivatives. To build this benchmark, we manually selected 10 widely-used pre-trained models (refer to Tbl. 4) with between 1 billion and 4 billion parameters (the upper bound was determined by our GPU memory constraints). Among these, we purposefully included four pairs of architecturally similar models from Meta, Microsoft, Google, and AliBaba to evaluate our tester's ability to distinguish between closely related base models and to have some control models. For each pre-trained model, we then randomly sampled 10 fine-tuned derivatives using the Hugging Face API (i.e. use highly reliable ground truth verification), prioritizing diversity in model creators. This sampling strategy resulted in 100 derived models, that constitute BENCH-A.

Table 4: All 10 pre-trained LLMs from BENCH-A (left) and top 10 from BENCH-B (right).

| Hugging Face Model | # params |
|---|---|
| meta-llama/Llama-3.2-1B-Instruct | 1,235,814,400 |
| meta-llama/Llama-3.2-3B-Instruct | 3,212,749,824 |
| microsoft/Phi-3-mini-4k-instruct | 3,821,079,552 |
| microsoft/phi-2 | 2,779,683,840 |
| google/gemma-2b | 2,506,172,416 |
| google/gemma-2-2b | 2,614,341,888 |
| Qwen/Qwen2-1.5B | 1,543,714,304 |
| Qwen/Qwen2.5-1.5B-Instruct | 1,543,714,304 |
| deepseek-ai/deepseek-coder-1.3b-base | 1,346,471,936 |
| TinyLlama/TinyLlama-1.1B-Chat-v1.0 | 1,100,048,384 |

| Hugging Face Model | # params |
|---|---|
| openai-community/gpt2 | 124,439,808 |
| EleutherAI/pythia-70m | 70,426,624 |
| microsoft/DialoGPT-medium | 345,000,000 |
| facebook/opt-125m | 125,239,296 |
| distilbert/distilgpt2 | 81,912,576 |
| openai-community/gpt2-large | 774,030,080 |
| openai-community/gpt2-medium | 354,823,168 |
| Qwen/Qwen2-0.5B | 494,032,768 |
| JackFram/llama-68m | 68,030,208 |
| EleutherAI/gpt-neo-125m | 125,198,592 |
| ... | ... |

Table 5: Domain-specific Hugging Face models included in our analysis.

| Domain | Model ID | Hugging Face URL |
|---|---|---|
| Medical | medical_transcription_generator | https://huggingface.co/alibidaran/medical_transcription_generator |
| Medical | gpt2-large-medical | https://huggingface.co/Locutusque/gpt2-large-medical |
| Medical | healthbot | https://huggingface.co/Anjanams04/healthbot |
| Law | bloom-560m-finetuned-fraud | https://huggingface.co/jslin09/bloom-560m-finetuned-fraud |
| Financial | FinguAI-Chat-v1 | https://huggingface.co/FINGU-AI/FinguAI-Chat-v1 |
| Financial | FinOPT-Lincoln | https://huggingface.co/MayaPH/FinOPT-Lincoln |
| Biology | distilgpt2-finetuned-microbiology | https://huggingface.co/as-cle-bert/distilgpt2-finetuned-microbiology |
| Astrology | astroGPT | https://huggingface.co/stevhliu/astroGPT |
| Game | magic-the-gathering | https://huggingface.co/minimaxir/magic-the-gathering |
| Game | chessgpt2-medium-l | https://huggingface.co/dakwi/chessgpt2-medium-l |

The second benchmark, denoted as BENCH-B, was constructed through a more automated and comprehensive approach. We began by downloading the 1,000 most popular models from Hugging Face with less than 1B parameters, ranked by download count. We then filtered out non-English models[3] and those exhibiting low entropy or high self-perplexity, which are indicators of poor training quality or insufficient learning[4]. This filtering process resulted in 608 viable models. To establish ground truth provenance relationships among these models, besides the model owners provided `fine-tune` keyword approach, we also used the other less reliable methods. Through this analysis, we identified 57 pre-trained models and established 383 ground-truth model provenance pairs. The remaining 148 models are considered to be independent, having no clear derivation relationship with any other models in analyzed set. Part of models from BENCH-B is given in Table 4.

**Examples of Candidates' Domains from Bench-A/B.** We find examples of candidates for domains such as financial, biology and medical among others. See Table 5 for a list of examples.

**Experimental Setup.** We run our model provenance testers on a Linux machine with 64-bit Ubuntu 22.04.3 LTS, 128GB RAM and 2x 24 CPU AMD EPYC 7443P @1.50GHz and 4x NVIDIA A40 GPUs with 48GB RAM. All experiments are implemented using PyTorch framework [35] and the Hugging Face Transformers library [46].

# D  Sampling Prompts

To produce prompts for our provenance testers, we use indiscriminately five popular LLMs: `gemini-pro-1.5`, `claude-3.5-sonnet`, `gemini-flash-1.5`, `deepseek-chat`, and `gpt-4o-mini`. Each produced prompt is an incomplete sentence containing five to twenty words – refer to Table 6 for examples.

Table 6: Examples of prompts.

```
1  In response to mounting public pressure, the concerned
2  The bright star known as Antares was visible even from
3  The surgeon prepared the instruments for a delicate
4  Scholars carefully examined the fragile
5  The phonetics lecturer explained the intricacies of the
```

---

[3]Due to lack of control models for them.

[4]We avoid testing low quality models.

| run | BENCH-A | | BENCH-B | |
|---|---|---|---|---|
| | precision | recall | precision | recall |
| 1 | 1.00 | 0.83 | 0.93 | 0.67 |
| 2 | 0.99 | 0.83 | 0.94 | 0.68 |
| 3 | 0.98 | 0.86 | 0.95 | 0.67 |
| 4 | 1.00 | 0.83 | 0.95 | 0.67 |
| 5 | 1.00 | 0.83 | 0.94 | 0.66 |

Figure 6: Precision and recall of the provenance tester on BENCH-A and BENCH-B with five different sets of 1,000 prompts.

| Model 1 | Model 2 | $k = 1$ | $k = 4$ | $k = 16$ | $k = 64$ |
|---|---|---|---|---|---|
| `gpt2-large` | `gpt2-medium` | 0.64 | 0.36 | 0.22 | 0.16 |
| `gpt2-large` | `megatron-gpt2-345m` | 0.64 | 0.38 | 0.25 | 0.15 |
| `pythia-410m-deduped` | `pythia-410m` | 0.62 | 0.37 | 0.24 | 0.15 |
| `gpt2-medium` | `megatron-gpt2-345m` | 0.62 | 0.34 | 0.22 | 0.15 |
| `Qwen1.5-0.5B` | `Sailor-0.5B` | 0.61 | 0.35 | 0.20 | 0.17 |
| ... | ... | ... | ... | ... | ... |
| | average | 0.33 | 0.13 | 0.08 | 0.06 |

Figure 7: Most similar pre-trained models from BENCH-B sorted for $k = 1$ (no advanced prompt sampling), and their corresponding values for $k = 4, 16, 64$.

# E   The Effectiveness of Rejection sampling

Certain pre-trained models from BENCH-A and BENCH-B exhibit a high degree of similarity when comparing their output tokens generated from random prompts. Table 7 presents the top 5 most similar model pairs from BENCH-B, measured by the percentage of matching output tokens when tested on 1,000 random prompts.

Table 7 demonstrates how the percentage of matching tokens changes with rejection sampling (columns $k = 4, 16,$ and $64$). For example, the most similar pair of models shows a reduction in matching output tokens from $64\%$ ($k = 1$) to merely $16\%$ ($k = 64$), indicating that rejection sampling significantly reduces token overlap between models. This improvement directly enhances the efficiency of provenance testing by reducing the tester's online complexity.

# F   Testing with Known Parent

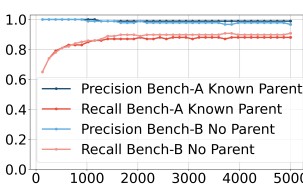 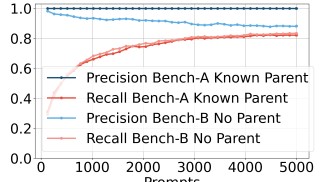 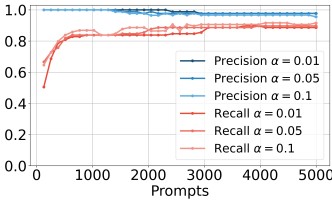

Figure 8: Precision/recall the tester on BENCH-A (left) and BENCH-B (right) without known parent (dark blue and dark red), and with a suspected parent (light blue and light red) ).

Figure 9: Precision and recall of running the model provenance tester on BENCH-A with different significance levels $\alpha$.

We evaluate the provenance tester for cases where the parent model is given, addressing whether a pair of models $(P, C)$ constitutes a provenance pair.

We construct test pairs from both BENCH-A and BENCH-B benchmarks. From BENCH-A, we take all 100 true pairs $(P_i, C_i)$ and create 100 false pairs $(\tilde{P}_i, C_i)$ by selecting one random non-parent $\tilde{P}_i \neq P_i$ for each child $C_i$. This ensures a balanced dataset where random guessing would achieve $50\%$ accuracy. We similarly obtain 766 testing pairs from BENCH-B[5].

Results from testing both benchmarks are shown in Figure 8. As expected, the tester performs better when the suspected parent is known compared to cases with unspecified parents. While recall remains unchanged, precision reaches 100%. This improvement occurs because false positives now require both that the statistical hypothesis test returns the wrong parent and that this wrong parent matches the suspected parent. The precision with known parents ($P_{knownparent}$) is lower

---

[5]Unlike BENCH-A, BENCH-B already contains (child) models that have no known parent among the 57 pre-trained models, and we use these as one of the negative pairs.

bounded by the precision with unspecified parents ($P_{unspecified}$), and depends on how we sample the incorrect parents. With uniform random sampling, it can be estimated as $P_{knownparent} = 100 - \frac{100 - P_{unspecified}}{n-1}$, where $n$ is the number of pre-trained models. For both benchmarks, this yields precision slightly below 100%.

## G   Precision/Recall for Different $\alpha$

We also run the model provenance tester for BENCH-A with different significance level $\alpha \in \{0.01, 0.05, 0.1\}$. The results of these experiments are given in Figure 9. As evident from the results, it is clear that the significance level does not have any major impact on the efficiency of the tester.

## H   Advanced Sampling of Prompts

We further give the rejection sampling of prompts used to reduce the online complexity of the tester. We present only the sampling procedure, i.e. how a set of prompts $x_1, \ldots, x_{T'}$ is produced given the original prompt sampler $\Omega$, and the two sets of models: the parent candidate set $F$, and the control set $C$. The tester that uses the new set is identical to the original tester, with the only difference that it samples prompts from the new set. The sampling is given in Algorithm 3.

---

**Algorithm 3** Advanced Prompt Sampling

---

**Require:** Parameter $k$, Candidate set $F = \{f_1, \ldots, f_s\}$, set of control models $C = \{c_1, \ldots, c_m\}$, prompt space $\Omega$, number of prompts $T'$

    $Prompts \leftarrow \emptyset$                                                       ▷ Initialize empty set of prompts

    $H \leftarrow F \cup C$                                                                 ▷ All models

    $same[i][j] \leftarrow 0$ for all $i, j \in [|H|]$     ▷ Counter of times two models produced the same output token

    **for** $i \leftarrow 1$ to $T'$ **do**

        $x_1, \ldots, x_k \overset{\text{iid}}{\sim} \Omega$                                           ▷ Sample $k$ prompts

        $Score[j] \leftarrow 0$ for all $j \leftarrow 1$ to $k$                 ▷ Scores across $k$ prompts

        **for** $j \leftarrow 1$ to $k$ **do**                                  ▷ Find score for each

            $s \leftarrow 0$

            **for** $l_1 \leftarrow 1$ to $|H|$ **do**

                **for** $l_2 \leftarrow 1$ to $|H|$ **do**

                    $old \leftarrow \frac{same[l_1][l_2]}{i-1}$

                    $new \leftarrow \frac{same[l_1][l_2] + \mathbb{1}(h_{l_1}(x_j) = h_{l_2}(x_j))}{i}$

                    $weight \leftarrow e^{\tau \cdot (old - new)}$

                    $s \mathrel{+}= \mathbb{1}(old > new) \cdot weight$

                **end for**

            **end for**

            $Score[j] \leftarrow s$

        **end for**

    **end for**

    $l \leftarrow \arg\max_j Score[j]$                                      ▷ Find largest score

    $Prompts \leftarrow Prompts \cup \{x_l\}$                             ▷ Add that prompt

    **return** $Prompts$

---

## I   Reducing Offline Queries with Best Arm Identification

To reduce the offline queries to the parent and control models we replace the hypothesis tests with Best Arm Identification (BAI) algorithm (that provides as well theoretical guarantees on confidence). For practical purposes, in our implementation given in Algorithm 4 we use the BAI proposed in [15].

---

**Algorithm 4** Tester based on Best Arm Identification

---

**Require:** Model $g$, candidate set $F = \{f_1, \ldots, f_s\}$, set of control models $C = \{c_1, \ldots, c_m\}$,
    prompt space $\Omega$, number of prompts $T$, significance parameter $\alpha$, maximum average prompts per
    model $N$

    $M \leftarrow F \cup C$            ▷ Set of all models

    $U(t, \alpha) := \sqrt{\frac{\log(4t^2/\alpha)}{2t}}$            ▷ Confidence interval for BAI

    hits$[m] \leftarrow 0$ for all $m \in M$            ▷ # same tokens with $g$

    tots$[m] \leftarrow 0$ for all $m \in M$            ▷ # queried

    $A \leftarrow M$            ▷ Active set of models

    $t \leftarrow 0$

    **while** TRUE **do**

        $x \overset{\text{iid}}{\sim} \Omega$            ▷ Sample prompt

        $y_g \leftarrow g(x)$            ▷ Query $g$

        **for** $m \in A$ **do**            ▷ Update hits/tots

            $y_m \leftarrow m(x)$            ▷ query model $m$

            $hits[m] += \mathbb{1}(y_g, y_m)$            ▷ Update hit counters

            $tots[m] += 1$            ▷ Update total queries

        **end for**

        $\mu_{best} \leftarrow \max_{m \in M} \{\frac{\text{hits}[m]}{\text{tots}[m]}\}$            ▷ Find best $\mu$

        $u \leftarrow U(t, \alpha)$            ▷ Confidence radius

        **for** $m \in A$ **do**

            **if** $\mu_{best} - u > \frac{\text{hits}[m]}{\text{tots}[m]} + u$ **then**

                $A \leftarrow A \setminus \{m\}$            ▷ $\mu_m$ too far from best

            **end if**

        **end for**

        **if** $|A| = 1$ **then**            ▷ Only 1 model left

            **break**

        **end if**

        **if** $\sum_{m \in M} \text{tots}[m] > N \cdot |M|$ **then**            ▷ Reached max queries

            **break**

        **end if**

        $t \leftarrow t + 1$

    **end while**

    **if** $|A| = 1$ and $A \subseteq F$ **then**            ▷ Model needs to be from $F$

        **return** (TRUE, $A$)

    **end if**

    **return** FALSE

---

