# OpenReview forum: "Model Provenance Testing for Large Language Models"
_NeurIPS.cc/2025/Conference — NeurIPS 2025 poster_

### Official Review · Reviewer_CgZ4 · 2025-06-12

**Clarity:** 2
**Significance:** 3
**Originality:** 3
**Rating:** 5
**Confidence:** 3

**Summary:**

This paper presents a model provenance testing framework tailored for large language models (LLMs). The core insight is to infer derivational relationships between models by quantifying output discrepancies across different models when presented with identical prompts. The authors construct two test benches of varying scales, collecting divergence data from the outputs of a target model $g$, a base model $f$, and a control model ensemble. Leveraging statistical hypothesis testing, the framework analyzes the significance of $g$ being a derivative of $f$, so as to realize the detection of model provenance. Experiments show that the tester achieves a precision of 90%-95% and a recall of 80%-90% in identifying derived models, proving the feasibility of systematic provenance verification through only API access.

**Questions:**

1. The sizes the models selected in the paper are all below 4B. Then, is this method applicable to models which sizes are above 4B?

2. Model distillation has become an important means of model derivation. Distillation can lead to a significant compression of the model size. Considering that the method proposed in this paper mainly uses models with similar parameter scales, is this method also applicable in the scenario of model distillation?

**Ethical Concerns:**

["NO or VERY MINOR ethics concerns only"]

**Final Justification:**

Thanks for the authors' rebuttal. They have done a great job in model provenance, using a novel method. I agree with their viewpoint and will update my score from "2: borderline reject" to "5: accept".

**Limitations:**

yes

**Quality:**

3

**Strengths And Weaknesses:**

**Strengths**

1. This paper presents a model provenance testing method based on multiple hypothesis test.

2. If the weaknesses mentioned below can be clarified, the method described in this paper can strongly prove that Assumption 1 holds true， and it can also be considered that the Bench constructed in the paper has the potential to become a Benchmark for detecting the derivative relationships of the model.


**Weakness**

1. It may be flawed in this paper to solely focus on selecting the **next token** when calculating the similarity between the outputs of two LLMs. For instance, when the **next token** is an adverb, pronoun, or other function word, this approach may compromise the validity of the similarity calculation. The author(s) should provide a clear elaboration on how such scenarios were addressed.

2. The paper neither delineates the construction principles for candidate provenance pairs and control sets nor furnishes a comprehensive inventory of Benches or analyzes the distribution of models within the Bench. A critical oversight arises in the potential bias introduced during control set selection—for example, disproportionately including derivative models of a single pre-trained model. The author(s) should clarify and explain in this aspect.

3. The construction of the benchs is relatively simple. For example, in Bench A, 100% of the models are derived models. If I construct a simple model that returns true every time, I can obtain 100% precision and recall. For Bench B, my model can achieve a precision of nearly 80% and a recall of 100%. It seems that the performance of my model is close to the method proposed in this paper.

4. This paper primarily demonstrates the effectiveness of the detection method in model provenance testing via precision and recall metrics. The underlying concept appears analogous to Instance-based clustering or anomaly detection. If so, the methodology could be strengthened by further detailing how multiple hypothesis testing results guide parameter selection for clustering or anomaly detection algorithms. Alternatively, presenting the clustering outcomes of known model provenance relationships using this method would offer a visual means to validate the approach’s efficacy from a complementary perspective.

**Minors**

1. The two sub-figs in Figure 3 are arranged vertically. The descriptions in the caption is explained in left-right. It is suggested to make adjustments and modifications.

---

> ### Author Rebuttal · Authors · 2025-07-31
>
> We thank all reviewers for their time and constructive feedback. Based on the feedback, we will revise the paper to clarify points raised by the reviewers.
> Below, we first provide a general response to outline the scope and goals of our work and then respond to each reviewer's specific points in the order we have received them.
>
>
> # General Response
>
> Our work addresses the challenge of model provenance testing, a practical problem formulation that, to the best of our knowledge, has not been systematically analyzed before. We introduce the problem and provide a solution in one of the most practical (and constrained) settings: requiring only black-box API access to models that return just the next predicted token.
>
> Our goal was to propose and empirically validate an initial framework for this new problem. However, we recognize that the problem space is extensive, with many potential variations. A comprehensive analysis of all these variations was beyond the scope of this work, which is the first to establish the viability of the core approach. The reviewers'  questions—for instance, regarding alternative similarity metrics, different derivation techniques, and more complex threat models—highlight several of these valuable and important directions for future investigation. Our approach is extensible and we believe it provides a solid starting point for future work.
>
> # Response to Reviewer CgZ4
>
> * **On the next-token similarity metric (Weakness 1):** While individual tokens can be noisy, our approach relies on statistical aggregation across thousands of diverse prompts, which reveals a robust signal of provenance.
>
> * **On benchmark construction and control sets (Weakness 2):** We will release the full list of models used in our benchmarks for transparency to show the lack of large bias. Moreover, BENCH-A has perfectly balanced parent sets, as each of the 10 pre-trained parent models has exactly 10 fine-tuned children models, thus ensuring no bias in the parent selection.
>
> * **On the perceived simplicity of the benchmarks (Weakness 3):** This appears to be a misunderstanding of our evaluation. We do not evaluate a simple binary classifier for each pair. Rather, our tester (according to Algorithm 1) must identify one single correct parent from a combined set of candidate parents. If it cannot do that conclusively with statistical error rate set, it will output that there is no parent. Thus the baseline precision for BENCH-A is 1/(10+1) = 9%, and for BENCH-B it is 1/(57+1) = 1.7%, not as high as stated in the review.
>
> * **On the analogy to clustering (Weakness 4):** While conceptually similar to anomaly detection, our method provides a more principled framework. Instead of relying on an arbitrary similarity threshold, we use a dynamic baseline of control models to determine what constitutes a statistically significant similarity. This, combined with the formal control of the Family-Wise Error Rate, is a key advantage and contribution of our work.
> A clustering approach doesn’t provide any such statistical guarantees, and our insights in the problem formulation about using control models is complementary to these other techniques.
>
> * **On applicability to models > 4B (Question 1):** The 4B parameter limit was imposed by our available hardware platform. The statistical principles of our tester are model-size agnostic, and we expect the method to generalize to larger models, as the underlying similarity patterns are unlikely to disappear.
>
> * **On model distillation (Question 2):** We have not evaluated our method against model distillation. As the first paper to formalize provenance testing in this setting, our scope was focused on fine-tuning, a common derivation technique. Investigating other derivation methods like distillation can be a direction for future work to explore.

---

> > ### Comment · Reviewer_CgZ4 · 2025-08-05
> >
> > I still have reservations about using only the selection of the next token as a criterion for evaluation. Currently, large models themselves take into account the diversity of generated content, and the next tokens obtained by submitting the same prompt to a large model multiple times will inherently vary. Such situations may be less common in models with a parameter scale of less than 4B, but this also limits the applicability of the method proposed in this paper. Additionally, although the method in this paper can probabilistically indicate that two models with similar output distributions have a derivative relationship, it cannot effectively prove that two models with different output distributions have no derivative relationship. This still weakens its supporting for model provenance identification.

---

> > > ### Author Response · Authors · 2025-08-07
> > >
> > > Thank you for the response to our rebuttal. We would like to address the remaining concerns below.
> > >
> > > 1.  The reviewer highlights a concern that diversity (e.g., via setting different temperature values in generation) affects the variability of text output even under the same prompt. This is known and we do not disagree. For instance, an EMNLP 2024 paper [1] finds differences in output for different values of temperature. However, these outputs are still similar under other metrics, e.g., [1] considers similarity of text in terms of Jaccard, Levenshtein, Bag-of-Words metrics which remains high. The observation generalizes over other larger models (e.g. Fig 6,7,8 in [1] have larger than 4B parametrs). Temperature setting also depends on the application domain as higher temperatures can impact task performance.
> > >
> > >
> > >       Notwithstanding, the central framework of Holm–Bonferroni with control model that we’ve proposed can be used as-is with these other metrics since the similarity across multiple runs is high even for larger models when diversity is used.
> > >
> > >
> > >
> > > 2. The reviewer highlights that our proposed scheme does not constitute a proof of provenance. We point out that our methods already give an upper bound on the statistical error (i.e., FWER). Most works that test black-box models for other properties, e.g. data taggants [2], membership inference [3], property inference [4], all have to use similar statistical error bounds. The FWER upper bound our framework gives is equivalent to the FP upper bound that hypothesis testing gives (the Neyman Pearson lemma).
> > >
> > >
> > > [1] Renze, Matthew. "The effect of sampling temperature on problem solving in large language models." Findings of the association for computational linguistics: EMNLP 2024. 2024.
> > >
> > > [2] Bouaziz, Wassim, El-Mahdi El-Mhamdi, and Nicolas Usunier. "Data taggants: Dataset ownership verification via harmless targeted data poisoning." arXiv preprint arXiv:2410.09101 (2024).
> > >
> > > [3] Carlini, Nicholas, et al. "Membership inference attacks from first principles." 2022 IEEE Symposium on Security and Privacy (SP). IEEE, 2022.
> > >
> > > [4] Wang, Lijin, et al. "Property existence inference against generative models." 33rd USENIX Security Symposium (USENIX Security 24). 2024.

---

### Official Review · Reviewer_ckdW · 2025-06-30

**Clarity:** 3
**Significance:** 3
**Originality:** 3
**Rating:** 5
**Confidence:** 3

**Summary:**

The paper proposes a statistically rigorous test for LLM provenance: that is, given black-box (API) access to a target model and a set of candidate parent models and controls, it determines whether the target model is a fine-tune of one of the candidate parents. The procedure asks whether the first-token agreement rate between the target model and the proposed parent is statistically greater than the agreement rate between the target model and each of a set of control models, over a randomly sampled set of prompts, using a one-sided two-proportion Z-test. Since a test is run against each candidate parent, Bonferroni correction is used to control the FWER. Experiments on multiple benchmarks over more than 600 public models achieve 90–95 % precision and 80–90 % recall with 3,000 prompts. Entropy‑based prompt rejection sampling drops query cost by up to 6x with no reduction in accuracy.

**Questions:**

- Can this method be extended to consider entire model outputs rather than just the first token? Using only the first token means we're unable to take advantage of differences in style or phrasing between models.

- How do you choose the control models? Are there guidelines for doing so? For example, should controls cover a range of architectures, parameter counts, or training data domains?

**Ethical Concerns:**

["NO or VERY MINOR ethics concerns only"]

**Final Justification:**

My concerns have been addressed by the rebuttal. Thank you to the authors for taking the time to reply!

**Limitations:**

Prompts $x_1, \cdots, x_T$ are shared across models, which creates a statistical dependence between the two proportions (parent-target and control-target) used in each Z-Test. Does this violate the test’s variance calculation?

**Paper Formatting Concerns:**

None.

**Quality:**

3

**Strengths And Weaknesses:**

Strengths:

- The test is simple and elegant, requires only black‑box access to models, and is original: it is the first technique that offers formal FWER guarantees for model‐provenance claims. The additional entropy‑based prompt rejection is a novel and interesting idea.

- the authors release two large, well‑curated benchmarks. Their evaluation is rigorous and comprehensive, and well-explained (RQ1 - 4). They run ablations on prompt count and control‑set size / selection.

Weaknesses:

- The test’s effectiveness depends on the choice of control models: if a control model happens to be very similar to the true parent (trained on overlapping data, from the same lineage, etc), the baseline agreement rate rises which could drop the power of the test, and the test may fail.

- Because only the first token is used, attackers could fine‑tune a model to match on that single token while diverging otherwise.

---

> ### Author Rebuttal · Authors · 2025-07-31
>
> We thank all reviewers for their time and constructive feedback. Based on the feedback, we will revise the paper to clarify points raised by the reviewers.
> Below, we first provide a general response to outline the scope and goals of our work and then respond to each reviewer's specific points in the order we have received them.
>
>
> # General Response
>
> Our work addresses the challenge of model provenance testing, a practical problem formulation that, to the best of our knowledge, has not been systematically analyzed before. We introduce the problem and provide a solution in one of the most practical (and constrained) settings: requiring only black-box API access to models that return just the next predicted token.
>
> Our goal was to propose and empirically validate an initial framework for this new problem. However, we recognize that the problem space is extensive, with many potential variations. A comprehensive analysis of all these variations was beyond the scope of this work, which is the first to establish the viability of the core approach. The reviewers'  questions—for instance, regarding alternative similarity metrics, different derivation techniques, and more complex threat models—highlight several of these valuable and important directions for future investigation. Our approach is extensible and we believe it provides a solid starting point for future work.
>
> # Response to Reviews ckdW
>
> * **On extending the similarity metric (Weakness 2, Question 1):** Our framework is metric-agnostic. We chose next-token agreement as it is the simplest available signal, and our results show it is surprisingly effective. The core methodology—comparing a candidate parent's similarity against a baseline of control models using hypothesis testing—can be directly applied to more complex metrics, such as n-gram overlap or semantic similarity of entire outputs, but will change the sample complexity therein.
>
> * **On selecting control models (Weakness 1, Question 2):** We adopted a simple and unbiased selection strategy in our experiments (the control set consists of all available pre-trained models), to show that even without manual curation of control models, the framework can still effectively identify model provenance. Potentially more robust selection would be to use a set of pre-trained models that are similar in domain to the target model. For instance,  if the target is a French-language model for generating Python code, good control models would include other French language models, Python code generation models, and ideally other French Python code generation models.
>
> * **On statistical dependence (Limitation):** The prompts are shared, but they are sampled uncorrelated with (independently of) the model pair being tested. This ensures that the set of prompts does not favor any particular parent or control model, and the independence required for the variance calculations thus holds.

---

> ### Author Response · Authors · 2025-08-07
>
> We thank the reviewer again for their feedback, and we are happy to further clarify any of the rebuttal points.

---

### Official Review · Reviewer_ExqM · 2025-07-01

**Clarity:** 4
**Significance:** 4
**Originality:** 4
**Rating:** 6
**Confidence:** 4

**Summary:**

This paper presents a framework for testing the provenance of large language models (LLMs). The motivation for this work stems from rising concerns about issues such as model stealing and backdoor vulnerabilities found in LLMs. The proposed method operates without any assumptions related to the training procedures or data, requiring only black-box access to the models.

The method is based on the empirical observation that fine-tuned models typically have output distributions more similar to those of their parent models than to unrelated ones. The authors employ statistical hypothesis testing to infer model provenance and validate their hypothesis. The method has been evaluated on a diverse set of LLMs from Hugging Face, ranging from 30 million to 4 billion parameters, achieving over 90% precision and over 80% recall. This represents the first systematic and practical approach to model provenance testing in constrained settings.

**Questions:**

Can we test with semantic similarity of the outputs a similar approach?

**Ethical Concerns:**

["NO or VERY MINOR ethics concerns only"]

**Final Justification:**

While the authors do not definitely solve the challenging problem they are presenting, I think the paper provides a great contribution on the field in that sense and is worth publishing.

**Limitations:**

The theoretical assumption that models will behave similar if derived from other is strong, but further testing on specific types of fine-tuning and post-training phases is needed.

**Paper Formatting Concerns:**

No formatting issues.

**Quality:**

4

**Strengths And Weaknesses:**

The problem has been very clearly defined. It is evident that this is a significant issue in the machine learning community, and the proposed solution addresses it to some extent. Additionally, I appreciate the authors' caution regarding false positive and false negative cases. This acknowledgment should alert the community to the possibility of such occurrences, prompting model owners to take action if they find that other models derived from theirs.

One of the strengths of this work is framing model provenance testing as a statistical hypothesis testing problem. The extensive evaluation provided also contributes significantly and novelly to the community.

The related work on watermarking and fingerprinting is accurate and highlights the novelty of the proposed approach. However, I would like to see more related work on the similarities in LLM outputs. While these studies may not focus on model provenance, several have compared LLM behavior across different datasets.

I have only a few minor concerns regarding this work. First, there could be a comparison with other baseline methods. For example, using output similarity (by measuring the similarity of embeddings) could help assess how well the method performs relative to simpler approaches.

Additionally, the initial hypothesis may have limitations depending on how the model has been fine-tuned. Further studies in the post-training stages could provide the community with more insights about the robustness of these methods.


In Summary, this is a novel and valuable contribution to the community, is well-conducted, and starts an open path for statistically asses model provenance.

---

> ### Author Rebuttal · Authors · 2025-07-31
>
> We thank all reviewers for their time and constructive feedback. Based on the feedback, we will revise the paper to clarify points raised by the reviewers.
> Below, we first provide a general response to outline the scope and goals of our work and then respond to each reviewer's specific points in the order we have received them.
>
>
> # General Response
>
> Our work addresses the challenge of model provenance testing, a practical problem formulation that, to the best of our knowledge, has not been systematically analyzed before. We introduce the problem and provide a solution in one of the most practical (and constrained) settings: requiring only black-box API access to models that return just the next predicted token.
>
> Our goal was to propose and empirically validate an initial framework for this new problem. However, we recognize that the problem space is extensive, with many potential variations. A comprehensive analysis of all these variations was beyond the scope of this work, which is the first to establish the viability of the core approach. The reviewers'  questions—for instance, regarding alternative similarity metrics, different derivation techniques, and more complex threat models—highlight several of these valuable and important directions for future investigation. Our approach is extensible and we believe it provides a solid starting point for future work.
>
> # Response to Reviewer ExqM
>
> Thank you for recognizing the novelty and value of our paper’s contribution.
>
> * **On related work:** As suggested, we can expand Section 2.1 to include recent studies that compare LLM output similarities, which will better contextualize our work.
>
> * **On baselines & semantic similarity (Question 1):** Our statistical framework is metric-agnostic. Using the semantic similarity of full outputs is a natural extension that can be directly integrated into our methodology. We agree this is a promising direction for future work to enhance robustness.
>
> * **On the limitations of the fine-tuning hypothesis:** While our work shows the similarity assumption holds for a wide variety of common fine-tuning methods, testing its boundaries against specific post-training techniques (e.g., distillation, extensive RLHF) is perhaps a next step in adapting to our proposed detection. Our paper provides the foundational methodology to enable such future studies.

---

> > ### Comment · Reviewer_ExqM · 2025-08-05
> >
> > I appreciate the answer of the authors answering my concerns. While I acknowledge there is work to be done, I think this paper presents a clear contribution to the field.

---

> > > ### Author Response · Authors · 2025-08-07
> > >
> > > We thank the reviewer again for their feedback and recognition of our paper’s contributions.

---

### Official Review · Reviewer_JAwj · 2025-07-03

**Clarity:** 2
**Significance:** 3
**Originality:** 2
**Rating:** 2
**Confidence:** 4

**Summary:**

This paper introduces a practical framework for model provenance testing in large language models (LLMs), under the realistic constraint of black-box access. The key insight is that fine-tuned models retain detectable output similarities to their parent models, which can be leveraged via hypothesis testing to establish provenance relationships. The method requires no knowledge of training data or architecture, and is benchmarked on over 600 real-world Hugging Face models (BENCH-A/B), achieving 90–95% precision and 80–90% recall. The authors also propose query-efficient variants using entropy-based prompt selection and bandit algorithms.

**Questions:**

1.How robust is the proposed method to adversarial fine-tuning or deliberate obfuscation aimed at evading provenance detection?
2.Can the authors provide a more formal theoretical characterization of their method? Specifically, what are the fundamental conditions under which the test is expected to succeed or fail? A clearer mathematical problem formulation would greatly strengthen the clarity and rigor.
3.Is there any provable regime (in terms of model divergence, prompt space, or number of queries) where the test is guaranteed to be accurate or, conversely, provably fails?
4.Could shared pretraining data or overlapping corpora (without actual fine-tuning) cause false positives? How does the method distinguish true derivation from incidental similarity due to data contamination or common training sources?

**Ethical Concerns:**

["NO or VERY MINOR ethics concerns only"]

**Final Justification:**

The rebuttal does not address core issues. Adversarial robustness remains untested beyond a non-adaptive threat model. Theoretical characterization is minimal—limited to standard FWER control without conditions or bounds for the core similarity assumption. Empirical scope is narrow, relying solely on next-token prediction despite acknowledged sensitivity loss for richer outputs. I maintain my reject recommendation.

**Limitations:**

Incorporating a theoretical analysis of the method's effectiveness would, in my view, substantially strengthen the novelty and contribution of this work.

**Quality:**

2

**Strengths And Weaknesses:**

Strengths

1.This paper is motivated by a timely and practical problem: how to determine whether an LLM is derived from another in realistic post-deployment settings where only API access is available.
2.The authors conduct extensive empirical validation on two benchmarks with over 600 models, covering a wide range of sizes and domains. Precision and recall results are strong and reported with appropriate statistical care.
3.Limitations and assumptions are clearly discussed; error sources are empirically analyzed.

Weaknesses

1.This paper assumes that fine-tuned models remain highly similar to parents, which may not hold under heavy domain shifts or adversarial training.
2.This paper lacks a theoretical guarantee on the effectiveness of the method beyond standard hypothesis testing; success depends on empirical regularities.
3.The control model selection is handled via simple random inclusion from popular models, which works in aggregate, but may struggle in edge cases (e.g., specialized domains).
4.The method currently focuses only on next-token prediction and ignores richer output structures, such as full generations or probability distributions. This limits sensitivity, especially for models trained to behave similarly on surface form but differ semantically.

---

> ### Author Rebuttal · Authors · 2025-07-31
>
> We thank all reviewers for their time and constructive feedback. Based on the feedback, we will revise the paper to clarify points raised by the reviewers.
> Below, we first provide a general response to outline the scope and goals of our work and then respond to each reviewer's specific points in the order we have received them.
>
>
> # General Response
>
> Our work addresses the challenge of model provenance testing, a practical problem formulation that, to the best of our knowledge, has not been systematically analyzed before. We introduce the problem and provide a solution in one of the most practical (and constrained) settings: requiring only black-box API access to models that return just the next predicted token.
>
> Our goal was to propose and empirically validate an initial framework for this new problem. However, we recognize that the problem space is extensive, with many potential variations. A comprehensive analysis of all these variations was beyond the scope of this work, which is the first to establish the viability of the core approach. The reviewers'  questions—for instance, regarding alternative similarity metrics, different derivation techniques, and more complex threat models—highlight several of these valuable and important directions for future investigation. Our approach is extensible and we believe it provides a solid starting point for future work.
>
>
> # Response to Reviewer JAwj
>
> * **On adversarial robustness (Weakness 1, Q1):** Our work considers a non-adaptive adversary. As the first paper to systematically address model provenance with such minimal assumptions, our focus is necessarily on this foundational threat model. While investigating adaptive attacks is an interesting direction for future work, it remains unclear if they could succeed without significantly degrading model performance, as we discuss in our submission (lines 76-80).
>
> * **On theoretical guarantees (Weakness 2, Q2, Q3):** Our method's theoretical guarantee is the formal statistical control of the Family-Wise Error Rate (FWER) via the Holm-Bonferroni procedure [16]. This is a standard and a robust result from statistical testing. Our central assumption—that derivation implies statistically significant similarity—is then validated empirically in Section 4.2, where we show it holds for the vast majority of cases in our large-scale benchmarks. Therefore, our tester’s accuracy is backed by formal statistical guarantees under an assumption that we demonstrate in real-world settings.
>
> * **On control model Selection (Weakness 3):** Our selection of control models was intentionally designed to be simple and unbiased (using all available top pre-trained models) to test the framework's general robustness and avoid introducing selection bias. The high performance on the diverse BENCH-B, which contains a wide variety of models (in total 57), demonstrates this approach works well without needing domain-specific manual curation for control models. Our two benchmarks have 600+ candidates, so manual curation would not have been feasible. Due to the large benchmark size, we expect to extensively cover the fine-tuning setup where a model is trained for specialized domains on custom datasets. For instance, we find examples of candidates for domains such as financial, medical and more (see below for a sample). We have not evaluated other selection strategies, which remains a direction for future work. We plan to release our benchmarks which we hope will create a baseline for future model provenance testing approaches.
> Examples of candidates’ domains from Bench-A/B:
>    - (Medical) https://huggingface.co/alibidaran/medical_transcription_generator
>    - (Medical) https://huggingface.co/Locutusque/gpt2-large-medical
>    - (Medical) https://huggingface.co/Anjanams04/healthbot
>    - (Law) https://huggingface.co/jslin09/bloom-560m-finetuned-fraud
>    - (Financial) https://huggingface.co/FINGU-AI/FinguAI-Chat-v1
>    - (Financial) https://huggingface.co/MayaPH/FinOPT-Lincoln
>    - (Biology) https://huggingface.co/as-cle-bert/distilgpt2-finetuned-microbiology
>    - (Astrology) https://huggingface.co/stevhliu/astroGPT
>    - (Game) https://huggingface.co/minimaxir/magic-the-gathering
>    - (Game) https://huggingface.co/dakwi/chessgpt2-medium-l
>
>
> * **On using only next-token prediction (Weakness 4):** We focus on the next token as it represents the minimal signal available in our black-box setting. Our results demonstrate that even this minimal signal is sufficient for high-accuracy provenance testing. One can consider stronger signals like n-grams, at the expense of increasing the required sample complexity. For instance, in small-scale, unreported experiments, we observed that the similarity rate is drastically reduced (as the probability of a match on two consecutive tokens is much lower than on one), suggesting a significantly larger number of queries would be needed to obtain statistically significant results.
>
> * **On distinguishing provenance from shared training data (Question 4):** This is a challenge as the models' exact training datasets are unknown. However, our benchmarks implicitly test this scenario. For instance, they include distinct models from the same family (e.g., google/gemma-2b and google/gemma-2-2b as well as a few other pairs) that are expected to have highly similar training datasets. The tester's high precision demonstrates its ability to distinguish the strong statistical signal of fine-tuning from the weaker background similarity arising from a shared datasets.
>
> [16] Sture Holm. A simple sequentially rejective multiple test procedure. Scandinavian journal of statistics, pages 65–70, 1979.

---

> > ### Comment · Reviewer_JAwj · 2025-08-08
> > **Post-Rebuttal Response**
> >
> > **Post-Rebuttal Response**
> >
> > The rebuttal does not resolve my main concerns:
> >
> > 1. **Adversarial Robustness Not Addressed** – The method is evaluated only under a non-adaptive threat model, with adaptive or adversarial scenarios deferred to “future work.” The claim that such attacks may degrade model performance is speculative and unsupported by experiments.
> >
> > 2. **Lack of Meaningful Theoretical Characterization** – The only “theoretical guarantee” cited is control of the Family-Wise Error Rate via Holm–Bonferroni, which is standard in hypothesis testing and unrelated to the central assumption that derivation implies detectable similarity. No formal conditions, bounds, or failure regimes are provided.
> >
> > 3. **Limited Scope of Empirical Validation** – The evaluation ignores richer output structures and relies solely on next-token prediction. The rebuttal concedes that more informative signals (e.g., n-grams) drastically reduce similarity but does not explore or quantify this trade-off, raising concerns about sensitivity in challenging settings.
> >
> >  Keep reject score.

---

> ### Author Response · Authors · 2025-08-07
>
> We thank the reviewer again for their feedback, and we are happy to further clarify any of the rebuttal points.

---

### Note · Authors · 2025-08-12

We thank the reviewers for the constructive discussion. We believe our work's primary contribution is establishing a new, critical area of research with a practical and robust foundational method:

* **Novel problem formulation in a realistic setting:** We are the first to systematically formulate LLM provenance testing and tackle under the most common real-world constraint: black-box API access. This setting is not a limitation, but a deliberate choice to ensure our method is immediately applicable in practice, where signals like output probabilities are often unavailable.

* **Principled and extensible framework:** The core of our contribution is the statistical framework itself—using control models to create a dynamic, statistically-grounded baseline. This framework is metric-agnostic. While we demonstrate its power with only the next-token signal, it can readily incorporate richer metrics (n-grams, semantic similarity) as valuable future extensions. Our work provides the foundation for this future research.

* **Large-scale empirical validation:** Our evaluation across 600+ real-world models is one of the largest of its kind. It provides strong empirical evidence that derivation leaves a statistically detectable signal, validating our core assumption and demonstrating our tester's high precision and recall (more than 80%-90%). We will release these benchmarks to stimulate further research.

Questions about adversarial robustness and other derivation methods are important, and they represent the next chapter of research into a problem our paper opens. We are confident we have provided a solid and essential foundation for this new problem.

---

### Decision · Program_Chairs · 2025-09-17

**Decision:**

Accept (poster)

**Comment:**

This work proposes a method for testing model provenance based on their token-level output distribution. The main contribution lies in a careful approach to hypothesis testing involving unrelated models to calibrate the alignment score. The resulting technique is able to detect whether one model is derived from another with high precision.

Reviewers generally agreed that this is a relevant problem and that the proposed technique, while straightforward, produces strong results across an extensive evaluation. Prominent concerns include the lack of adversarial robustness testing -- the work only considers non-adversarial settings -- and the relative simplicity of the next-token based analysis, which may perform poorly under more demanding settings (including adversarial settings).

Overall, this work presents a reasonably novel and thorough investigation of an effective approach, while not addressing some of the more technically complex topics in this space. In light of its novelty and to encourage further explorations in this space, we are recommending its acceptance.